# The Cysteine Protease Giardipain-1 from *Giardia duodenalis* Contributes to a Disruption of Intestinal Homeostasis

**DOI:** 10.3390/ijms232113649

**Published:** 2022-11-07

**Authors:** Rodrigo Quezada-Lázaro, Yessica Vázquez-Cobix, Rocío Fonseca-Liñán, Porfirio Nava, Daniel Dimitri Hernández-Cueto, Carlos Cedillo-Peláez, Yolanda López-Vidal, Sara Huerta-Yepez, M. Guadalupe Ortega-Pierres

**Affiliations:** 1Departamento de Genética y Biología Molecular, Centro de Investigación y de Estudios Avanzados del Instituto Politécnico Nacional, México City 07360, Mexico; 2Departamento de Fisiología, Biofísica y Neurociencias, Centro de Investigación y de Estudios Avanzados del Instituto Politécnico Nacional, Mexico City 07360, Mexico; 3Unidad de Investigación en Enfermedades Hemato-Oncológicas, Hospital Infantil de México Federico Gómez, Mexico City 06720, Mexico; 4Laboratorio de Inmunología Experimental, Torre de Investigación, Instituto Nacional de Pediatría, Mexico City 04530, Mexico; 5Programa de Inmunología Molecular Microbiana, Departamento de Microbiología y Parasitología Facultad de Medicina Universidad Nacional Autónoma de México, Mexico City 04510, Mexico

**Keywords:** giardipain-1, *Giardia duodenalis*, giardiasis, inflammation, pathogenesis, dysbiosis

## Abstract

In giardiasis, diarrhoea, dehydration, malabsorption, weight loss and/or chronic inflammation are indicative of epithelial barrier dysfunction. However, the pathogenesis of giardiasis is still enigmatic in many aspects. Here, we show evidence that a cysteine protease of *Giardia duodenalis* called giardipain-1, contributes to the pathogenesis of giardiasis induced by trophozoites of the WB strain. In an experimental system, we demonstrate that purified giardipain-1 induces apoptosis and extrusion of epithelial cells at the tips of the villi in infected jirds (*Meriones unguiculatus*). Moreover, jird infection with trophozoites expressing giardipain-1 resulted in intestinal epithelial damage, cellular infiltration, crypt hyperplasia, goblet cell hypertrophy and oedema. Pathological alterations were more pronounced when jirds were infected intragastrically with *Giardia* trophozoites that stably overexpress giardipain-1. Furthermore, *Giardia* colonization in jirds results in a chronic inflammation that could relate to the dysbiosis triggered by the protist. Taken together, these results reveal that giardipain-1 plays a key role in the pathogenesis of giardiasis.

## 1. Introduction

The intestinal epithelia of humans and animals are crucial for maintaining gut homeostasis and a balanced relationship between gut microbiota, digesta and the mucosal immune system. The disruption of this barrier can lead to a range of disorders, such as inflammatory bowel disease, colonic cancer and/or infectious diarrhoeal diseases [1,2].

Intestinal pathogens, including viruses, bacteria and parasites, often alter or disrupt gut epithelial homeostasis [3]. For instance, the pathophysiology of giardiasis involves intestinal disturbances frequently associated with changes in the epithelium. *Giardia duodenalis*, a protistan parasite that often causes enteritis (giardiasis) in humans and other animals is cross-transmissible among species (depending on genotype or assemblage). Infection is acquired via the accidental ingestion of drinking water or food contaminated with cysts, and trophozoites excyst. Although motile, many trophozoites attach to the tips of epithelial cells in the small intestine (usually duodenum and jejunum) where they replicate by binary fission [4]. Although infections with this parasite can be asymptomatic, many are not, and giardiasis ensues as a consequence of an imbalance in the intimate host-parasite relationship [5,6,7,8,9,10]. 

Despite decades of research, the pathogenesis of giardiasis is still not fully understood. However, we do know that *Giardia* produces a range of virulence factors including cysteine rich surface proteins, tenascins and cysteine proteases [5,10,11]. Virulence factors produced by *Giardia* can trigger a broad range of mechanisms to disrupt the homeostasis of the intestinal mucosa. For instance, some cysteine proteases appear to cleave the intestinal epithelial cytoskeletal protein villin and degrade cytokines including IL8 and CXCL1, CXCL2, CXCL3, CCL2 and CCL20 [12,13,14]. Other secreted proteases can disrupt intracellular junctions in intestinal epithelial cells, and trigger dysbiosis in the gut [14,15,16,17]. Recently, we discovered a cysteine protease, giardipain-1—GL50803_14019; prior designation: CP2 [18]—that is released by trophozoites and has gelatinolytic activity and pro-apoptotic effects [10]. The characteristics of giardipain-1 suggested that the protease could play a key role in the pathogenesis of giardiasis. For this reason, we undertook the present exploration to assess the effect(s) of giardipain-1 on the intestinal epithelium in the jird (*Meriones unguiculatus*) model [19] and investigated chaWB_nges in intestinal homeostasis and microbiota in jirds infected with trophozoites expressing giardipain-1. 

## 2. Results

### 2.1. Giardipain-1 Induces Apoptosis in the Jird Duodenum

Apoptosis induction by *G. duodenalis* compromises the epithelial barrier [10]. Therefore, we evaluated the effect of giardipain-1 on the induction of apoptosis using the intestinal loop model. As shown in Figure 1, Intestinal epithelial cells (IEC) undergoing apoptosis (Green) were detected in both control and giardipain-1-treated loops. However, treatment with giardipain-1 increased the number of apoptotic cells at 6 h p.i. (Figure 1H). Most apoptotic cells detected in giardipain-1-treated duodenal loops were epithelial cells that extruded from the surface of the villi, indicating that giardipain-1 triggers not only apoptosis but also anoikis in duodenal IEC (Figure 1D,H, white arrows). 

### 2.2. Giardipain-pAC-Transfected Trophozoites Enhance Epithelial Cell Damage in the Duodenum 

Next, we evaluated the role of giardiapain-1 in the establishment of giardiasis. To do this, jirds were infected with 2.5 × 106 WB trophozoites that had a low expression of giardipain-1 (WB^-^); with wild-type trophozoites (WB^+^) expressing giardipain-1; or with transfected trophozoites stably expressing giardipain-1 (WB giardipain-pAC trophozoites [10]. Uninfected jirds (n = 3 per date) receiving PBS were included as controls. Initially, we determined the expression of giardipain-1 in the three trophozoite lines used to infect the jirds, the WB giardipain-pAC, the WB^+^ and the WB^−^. In these assays, a 25 kDa band recognized by mAb 1G3 was detected at high abundance in giardipain-pAC trophozoites extracts, with a medium abundance in WB^+^ (wild type) and at a low level in WB^−^ (Figure 2A(a)). Loading control used was *G. duodenalis* α-tubulin (Figure 2A(b)). In the same extracts from WB^+^ and WB^−^ and WB giardipain-pAC, trophozoites, the 25 kDa band exhibited medium, low and high gelatinolytic activity, respectively (Figure 2B). 

However, an assessment of trophozoite burdens in jirds showed no marked difference among the groups infected with WB giardipain-pAC, WB^+^ and WB^−^ trophozoites at day 14, the time at which patency peaks in jirds [19,20,21,22] (Figure 3).

Plasmid detection was carried out in WB giardipain-pAC trophozoites cultured for for 15 to 21 days in the absence of puromicin and then for 10–20 days in the presence of this antibiotic to select the remaining trophozoites expressing the transfected plasmid. Cultures showed a positive growth demonstrating that indeed these trophozites still harboured the plasmid. Positive growth was also observed in axenized WB giardipain-pAC trophozoites recovered at days 7 and 21 p.i. from the intestine of jirds infected with WB giardipain-pAC trophozoites which were cultured in the presence of puromicin.

Next, we carried out histopathological examinations of the duodenal epithelia of jirds infected with either WB^−^, WB^+^ or WB giardipain-pAC trophozoites at 14, 21, 90 or 180 days. Jirds that received PBS showed normal mucosal and epithelial architecture. However, at 14 days p.i., the presence of *Giardia* trophozoites increased inflammatory cell infiltrate and thickened the villi. The inflammatory infiltration in the mucosa was represented mainly by lymphocytes and plasma cells. Enterocytic hyperplasia at the edge of the villi and histological alterations at the tips of villi were also observed in the presence of the trophozoites. Interestingly, these changes were more prominent in the mucosa of jirds exposed to WB giardipain-pAC (Figure 4M) as compared with those infected with WB^+^ trophozoites (Figure 4I). Less damage was evident in the duodenal epithelia from jirds infected with WB^−^ trophozoites which exhibited low girdipain-1 expression (Figure 4E). 

At 21 days p.i., the infiltration with lymphocytes and plasma cells persisted in the inflamed mucosa and that resulted in distortion of villi, characterized by a thickening due to inflammatory cell infiltration. Additionally, in some areas, the presence of a wider inter-villus space was observed, and a random shortening of the villi was also detected. Interestingly, distension and thickening of the tips of villi was associated with inflammatory cells and oedema, such that the villi appeared distorted. As for the observations made at 14 days, all of the histopathological changes were exacerbated when jirds were infected with WB pAC-giardipain-1 trophozoites (Figure 4N) compared with those infected with WB^+^ trophozoites (Figure 4J). As expected, WB^−^ trophozoites induced less damage in the duodenal epithelia than those infected with either WB giardipain-pAC or WB^+^ trophozoites (Figure 4F). 

At 3 months p.i., the multifocal separation, the shortening and the distortion of the villi clearly persisted. However, at this time point, we also detected fusion of villi (Figure 4G–I). The thickening of the villi observed in the infected jird was likely due to oedema in the submucosa at the tip of the villus and inflammatory cell infiltration at the crypt base. Again, the changes were more prominent in jirds infected with WB giardipain-pAC trophozoites (Figure 4O) compared with those infected with WB^+^ trophozoites (Figure 4K). Less damage was evident in the duodenal epithelia from jirds infected with WB^−^ trophozoites (Figure 4G).

At 6 months p.i., the alterations in the mucosa, including the distortion and villus fusion, thickening linked to the lymphoplasmacytic infiltration, shortening of the villi and oedema persisted in all trophozoites-infected groups. The changes were more prominent in jirds infected with WB giardipain-pAC trophozoites (Figure 4P) when compared with those infected with WB^+^ trophozoites (Figure 4L). Additionally, only in jirds infected with WB giardipain-pAC trophozoites (Figure 4P), we detected the presence of distended crypts and cell detritus (proteinaceous material and inflammatory cells) in the lumen that was compatible with exudative/suppurative cryptitis. There was less damage in the intestinal epithelium in jirds infected with WB^−^ trophozoites (Figure 4H) than those with WB giardipain-pAC or WB^+^ trophozoites, again indicating that giardipain-1 protein levels are important in the damage induced in the intestinal mucosa. 

Overall, these findings indicate that *Giardia* trophozoites induce significant morphological alterations in the intestinal mucosa of jirds. Furthermore, the histological scores corroborated that there was more mucosal damage in jirds infected with WB giardipain-pAC trophozoites than in those infected with WB^+^ or WB^−^ trophozoites (Appendix A). Further, the number of globet cells per villi was higher in jirds infected with WB giardipain-pAC trophozoites than in those infected with WB^+^ or WB^−^ trophozoites (Appendix A).

### 2.3. Trophozoites Increase Epithelial Cell Proliferation in the Duodenum

Histological damages are often related to changes in cell homeostasis. Therefore, we evaluated cell proliferation in the mucosa of jirds infected with WB^+^ or WB giardipain-pAC by detecting the presence of Ki67 and pHist3 [23]. Immunofluorescence staining of both proliferation markers was carried out in the duodenum of jirds infected with *Giardia* for 14 or 21 days or 3 months. As shown in Figure 5A,D,G,J,M,P, in the duodenum of control animals, Ki67 and pHist3 were detected in IEC at the crypt base, and the number of Ki67- and pHist3-positive cells remained constant, irrespective of the time at which the animal was euthanized. However, during giardiasis, Ki67-positive cells increased at the crypt base at 14 and 21 days p.i., irrespective of the line of trophozoites used (Figure 5B,C,E,F). By 3 months p.i., IEC Ki67-positive cells were only abundant in the duodenum of jirds infected with WB overexpressing giardipain-1 (Figure 5H,I). Similarly, to Ki67-positive cells, IEC pHist3-positive cells increased substantially in the duodenum of jirds infected with trophozoites at day 14 p.i., irrespective of the trophozoite line (Figure 5J–L). At 21 days or 3 months p.i., the number of pHist3-positive cells were marginally increased compared with those of control jirds (Figure 5M–R). Taken together, these results indicate that giardiasis increases proliferation of IEC.

### 2.4. Giardia Induces Goblet Cell Hyperplasia

Increased proliferation of IEC during infection is often associated with hyperplasia of goblet cells [24] and it is known that giardiasis increases the number of goblet cells in the small intestine [25]. Thus, we analyzed the presence of goblet cells in the intestine of jirds infected with WB^+^ or WB giardipain-pAC trophozoites. To this end, polysaccharides and mucus were analyzed by PAS-AB staining in the duodenum of jirds infected with trophozoites for 14 and 21 days and 3 and 6 months. As shown in Figure 6A–D in uninfected controls, the goblet cells were randomly distributed along the crypt-villus axis, and its numbers remained constant with age. In contrast, the number of goblet cells increased moderately in the duodenum of jirds infected with each trophozoite line, starting at 14 days p.i., but was significantly less in jirds infected with WB^−^ trophozoites (Figure 6E). The random distribution of goblet cells was maintained along the crypt axis in jirds infected with either WB^−^ or WB^+^ or WB giardipain-pAC trophozoites (Figure 6E,I,M). Starting at day 21 p.i., the number of goblet cells in the intestinal mucosa of jirds infected with WB^+^ and WB giardipain-pAC trophozoites increased significantly, (Figure 6F,J,N) and their presence was more prominent at the crypt base of the villi in the proliferative compartment. 

At 3 months p.i., WB^+^ trophozoites and WB giardipain-pAC trophozoites (Figure 6K,O) induced more goblet cells with larger mucus vacuoles in the intestinal crypts of duodenum when compared with duodenum crypts of jirds infected with WB^−^ trophozoites (Figure 6G). Interestingly, in infected jirds in general, the number of goblet cells at the apex of the villus was reduced. The numbers and distribution of the goblet cells in the intestinal mucosa of infected jirds was maintained for 6 months p.i. as compared with control jirds. However, when compare between the different trophozoite strains the number of goblet cells was higher when jirds were infected with WB trophozoites overexpressing pACgiardipain-1 (Figure 6P) than those infected with WB^+^ trophozoites (Figure 6L) or with WB^−^ trophozoites (Figure 6H). These findings show clearly that giardipain-1 significantly increases the number of goblet cells in the duodenum of infected jirds.

Subsequently, we explored the distribution of mucin 2 (MUC2), the main muco-polysaccharide produced by goblet cells in the intestine. In control jirds, MUC2 (green) was enriched in the cytosol of goblet cells randomly distributed along the crypt-villi axis (Figure 7A,D,G). The number of MUC2-positive cells along the crypt axis was constant, irrespective of jird age. Additionally, a faint layer of MUC2 on the apex and the epithelial brush border was readily detected (Figure 7A, arrow). However, the infection with WB^+^ or WB giardipain-pAC trophozoites for 14 and 21 days and 3 months markedly thickened the layer of MUC2 on the luminal side and increased the number of MUC2-positive cells (Figure 7B,C,E,F,H,I). The layer of MUC2 observed along the whole crypt-villi axis at 14 days p.i. suggested an active and relatively uniform secretion of the mucopolysaccharide by the goblet cells (Figure 7B,C). However, after 21 days and 3 months p.i., the layer of MUC2 was concentrated at the crypt base, suggesting an active secretion by newly-formed goblet cells and an atrophy or exhaustion of superficially-located goblet cells (Figure 7E,F,H,I). Importantly, at 3 months p.i., the luminal layer of MUC2 was more prominent in the duodenum of jirds infected with WB giardipain-pAC trophozoites than in jirds infected with WB^+^ trophozoites. Taken together, these findings indicate that IEC proliferation gives rise to ‘hypersecretory’ goblet cells during giardiasis. 

### 2.5. Giardia Triggers a Chronic Inflammatory Response in the Duodenum

The T-cell response triggered during giardiasis in the intestine can clear *Giardia* infection at around 6 weeks [26]. However, our results reveal a chronic inflammatory response in the duodenum of jirds infected with *Giardia* that extends beyond the 6-week window. Thus, we evaluated the presence of T cells in the duodenum mucosa of jirds infected with WB^+^ or WB giardipain-pAC trophozoites for 14 and 21 days and 3 months. CD3^+^ cells were detected in the lamina propria of the intestinal crypts, and these cells were in close apposition with the basal membrane of IEC (Figure 8A,D,G). As expected, the number of CD3^+^ cells increased markedly in the duodenum mucosa of jirds infected with WB^+^ or WB giardipain-pAC trophozoites for 14 and 21 days (Figure 8B,C,E,F). However, the number of CD3^+^ cells remained elevated in the duodenal mucosa of jirds at 3 months p.i. when the *Giardia* infection had already been cleared (Figure 8H,I). Importantly, in the duodenal mucosa of jirds infected with WB^+^ or WB giardipain-pAC trophozoites, CD3^+^ cells were detected in the lamina propria along the crypt-villi axis (Figure 8B,C,E,F,H,I), and these cells were enriched around the crypt base. Furthermore, a higher number of CD3^+^ cells were recruited into the lamina propria in jirds infected with WB giardipain-pAC trophozoites that those infected with WB^+^ trophozoites, suggesting a role for giardipain-1 in this process. These results indicate that giardiasis triggers a strong chronic immune response mediated by CD3^+^ cells in the duodenum of jirds.

### 2.6. Faecal Microbiota Transplanted to Jirds Infected with Giardia Triggers Epithelial Dysfunction and Changes in the Microbiome Composition 

Based on previous results [15,16], we tested the hypothesis that the immune response elicited by giardiasis is mediated by changes in the microbiome. We approached this by transplanting faecal microbiota from jirds infected with WB giardipain-pAC trophozoites to a group of jirds kept as recipient controls and to another group of jirds infected with WB giardipain-pAC trophozoites. Following transplantation of faecal microbiota, microscopic examination of duodenal sections was carried out at different time points. At 21 p.i., control jirds had a normal mucosal and intestinal epithelial structure (Figure 9A). However, the intestinal villi in jirds inoculated with microbiota from WB giardipain-pAC treated animals displayed villi thickening due to inflammatory infiltration (mainly lymphocytes) in the submucosa (Figure 9B). Additionally, when the microbiota was transplanted in the group of jirds infected with WB giardipain-pAC, the changes in the intestinal epithelium were exacerbated. In those jirds, the villi were shortened, distorted and fused; they were also thickened, likely due to infiltration of predominantly lymphocytes and plasma cells. This thickening extended to the base of the crypts and was consistent with crypt hyperplasia (Figure 9C). The histochemical analysis of duodenal sections obtained at 21 days p.i. from jirds inoculated with microbiota and then infected with WB giardipain-pAC trophozoites showed that goblet cells were more numerous, mainly in the crypts, contrasting lower numbers at the apices of villi (Figure 9C). In microbiota giardipain-pAC treated group, there was a slight increase in the number of positive cells in the crypts (Figure 9B) compared with the PBS-treated control group (Figure 9A). 

Alterations in the intestinal microbiota that associate with intestinal inflammation [27] were studied. To determine whether *Giardia* infection impacts the microbiome in jirds, DNA from faeces were analyzed using the Illumina Miseq platform. The results showed that, when faecal transplants were given to jirds, there was a reduction in relative abundance of members of the families Alcaligenaceae and S24-7, while in jirds that received the microbiota and were infected with WB giardipain-pAC trophozoites, there was a reduction of F16 and an increase in the relative abundance of members of the Enterococcaceae, and Aerococcaceae was detected (Figure 10A). Based on Bray–Curtis dissimilarities, no significant effect was observed on the bacteria community in faecal samples from jirds given microbiota or in jirds that received microbiota and had been infected with WB giardipain-pAC trophozoites (Figure 10B). However, the heatmap did show significant changes in the bacterial taxa (Figure 10C) between samples from jirds which received microbiota or those receiving microbiota and were infected with WB giardipain-pAC trophozoites. In the latter jirds, we observed a decrease in species of *Sutterella* (−3.45) and *Oscillospira* (−4.60). Interestingly, when this group of jirds was compared with the control group, a decrease in *Ruminoccoccus gnavus* (−6.23) and *Oscillospira* (−4.86) was observed. Thus, infection by *Giardia* trophozoites that overexpress giardipain-1 induce temporal disruption in the mucosa besides to impact in impaired microbiome eubiosis resulting in the inflammatory response observed in the infected animals.

## 3. Discussion

Here, we showed that giardipain-1 triggers apoptosis and an extrusion in epithelial cells (anoikis). This type of cell death occurs when epithelial cells lose intercellular contacts and undergo extrusion from the monolayer [28]. The present findings strongly support previous results indicating that giardipain-1 can induce apoptosis in epithelial cells [29,30] by targeting proteins at the tight and adherens junctions [10]. Low rates of apoptosis/anoikis in the intestinal epithelium eliminate aged or ravaged cells and maintain intestinal epithelial homeostasis [31]. Thus, our findings indicate that giardipain-1 affects epithelial homeostasis. The effect of giardiapain-1 is rapid and likely transient, given that in the intestinal mucosa the restoration/compensation mechanisms in the epithelium are activated immediately following infection, and is plausible that for that reason *Giardia* is rapidly eliminated from the gut. However, by destroying intercellular junctions and altering epithelial barrier functions, giardipain-1 might trigger the chronic effects that are associated with giardiasis, including the persistent pathological changes seen in the intestinal epithelium and/or the adaptive immune responses [15,32]. In fact, in our experiments the damage to the superficial mucosa of the small intestines of jirds infected with WB giardipain-pAC trophozoites was similar to that seen in people with celiac disease [33] or inflammatory bowel disease (IBS) [34]. Such damage includes the loss of normal villus structure, with a reduction in villus height, a marked enlargement and lengthening of crypts, and inflammatory cell infiltration with an increase density of intraepithelial lymphocytes [35]. Therefore, we propose that treatments aimed at inhibiting giardiapain-1 could help prevent various disorders associated with *Giardia duodenalis* infection, including IBS. 

The acute pathological changes associated with the trophozoites included increased lymphoplasmacytic infiltration in the villi, fused villi, and changes in the cytoarchitecture of the villi; many of these alterations extended into the chronic phase of the infection. In this context, the infection of jirds with trophozoites overexpressing giardipain-1 potentiated acute and chronic inflammatory effects induced by normal trophozoites, indicating that giardiapain-1 contributes markedly to the pathogenesis of giardiasis. Our results likely explain villus atrophy, intestinal crypt hyperplasia and increased numbers of immune cells in the mucosa of patients with giardiasis, observed in previous works [36,37,38,39]. 

An important observation in jirds infected with the different trophozoite lines was the increase in number and size of goblet cells and the presence of luminal mucus in the intestinal crypts. This observation accords with previous observations of goblet cell hyperplasia associated with infections with *Trichinella spiralis* [36,37], *Eimeria* [38], *Nippostrongylus brasilensis* [39,40], *Trichuris muris* [41,42] and *Giardia* [43,44,45]. Here, we speculate that mucus hypersecretion is a compensatory mechanism, aimed at repairing barrier function of the intestinal epithelium. However, this could also act as a defense mechanism given cysteine protease activity has been linked to mucin degradation and depletion, and the alteration of mucin gene expression in *Giardia* [46,47]. Given that giardipain-1 stimulates epithelial removal and proliferation, we cannot exclude the possibility that proteolytic processing of the intercellular junctions by giardipain-1 enhances epithelial cell turnover and leads to a disruption of the epithelial barrier. Overall, our results indicate that proteolytic activity mediated by giardipain-1 triggers epithelial plasticity in the gut. Further work is needed to investigate role of signaling pathways in the induction of goblet cell hyperplasia during giardiasis. 

Previous studies have shown that *Giardia* proteases can alter the composition of the intestinal microbiota and promote the formation of pathobionts [15,48,49]. Given that the intestinal microbiome protects the host against invasion by pathogens and against the overgrowth of pathobionts [50,51], this process is likely important in the pathogenesis of giardiasis. Indeed, the colonisation of pathobionts has been linked to the induction of inflammatory processes associated with the destruction of the epithelial barrier [52]. Further, it has been reported that the disruption of the microbiota during giardiasis can predispose to gastrointestinal disorders [15]. Interestingly, in our work, the alteration of bacterial taxa, such as a reduction in *Sutterella, Rimunococcus gnavus* and *Oscillospira,* in jirds infected with trophozoites, was similar to that observed in various inflammatory disorders of the small intestine [53,54,55]. On the other hand, the increase in members of the Enterococcaceae and Aerococcaceae may be due to alterations of the mucosal layer, together with changes in the innate and acquired immune responses, leading to an increase of these commensal microbiota as a consequence of dysbiosis in this model system. Their transcytosis through M cells, inducing IgA and IL 22 cytokines, as observed in mammalian hosts [56], could be a critical response to stimulate dendritic cells. Further, it has been reported that infection with the *Giardia duodenalis* strain GS causes systemic dysbiosis of commensal aerobic and anaerobic bacteria in the small intestine, specifically an increase in Proteobacteria and a decreased in Firmicutes and Melainabacteria [32], which can also be involved in post-infection complications associated with *Giardia* infection. Taken together, this information suggests that the microbiota play a key role during and after infection with *Giardia*, and that alterations in the microbiome could be linked to the development of some chronic inflammatory or immunological disorders [54]. 

In conclusion, giardipain-1-induced alterations seen in the intestinal epithelial cells and in the microbiome contribute to a better understanding of the pathogenesis of giardiasis and its associated complications.

## 4. Material and Methods

### 4.1. Parasite Culture

*Giardia duodenalis* trophozoites of the WB strain, (Assemblage A, ATCC# 30957), that were isolated from a patient (W.B.) with chronic giardiasis [57], were grown at 37 ºC in Keister’s modified TYI-S-33 medium [58] containing 10% bovine serum (HyClone, South Logan, UT, USA) with 1% antibiotic/antimycotic mixture (Code SV30079.01 HyClone, South Logan, UT, USA). WB giardipain-pAC trophozoites [10] were grown in the same medium, supplemented with 100 μM puromycin hydrochloride (Sigma-Aldrich; Merck KGaA, Darmstadt, Germany). Trophozoites were harvested and processed as previously reported [10]. 

### 4.2. Purification, Cloning and Expression of Giardipain-1

Cloning and expression of giardipain-1 in the WB giardipain–pAc line was described previously [10]. Giardipain-1 was purified by affinity chromatography using a monoclonal antibody (mAb IG3) coupled to Sepharose 4B [10]. 

### 4.3. Plasmid Detection in WB Giardipain-pAC Trophozoites by Puromycin-Selection 

WB giardipain pAC trophozoites were cultured in the absence of puromycin for 15 or 21 days, after which puromycin at 100 µM was added and parasite growth was determined over a period of one week. In vivo, WB giardipain pAC trophozoites were collected from the small intestines of infected jirds at days 7 and 21. Then, trophozoites obtained were axenised, cultured in the presence of puromycin at 100 µM and the trophozoite growth determined.

### 4.4. Assessing the Effect of Purified Giardipain-1 on Intestinal Loops

Male jirds (8–10 weeks of age), maintained in a purposely-designed laboratory using a standard day/night light cycle and provided with a commercial diet and water ad libitum. All experimentation was approved by the Internal Committee for Care and Use of Laboratory Animals guidelines (CICUAL). The intestinal loop model was as described previously [59]. Briefly intestinal loops of jirds we injected with 32 µg of purified giardipain-1 in 200 µL of Dulbecco´s Modified Eagle Medium (DMEM). Control animals received the same volume of DMEM only. Then apoptosis in the intestinal epithelium was determined at 4 and 6 h after giardipain-1 injection using the TUNEL assay [60].

### 4.5. Analysis of Giardipain-1 Expression in WB Giardipain-pAC, WB^+^ and WB^−^ Trophozoites by Western Blot, and Detection of Giardipain-1 Protease Activity by Zymogram Analysis 

WB giardipain-pAC, or WB^+^ or WB^−^ trophozoites (10 × 10^6^) were pelleted, suspended in 500 µL of PBS and sonicated in an ice bath at 60% amplitude. Three pulses were initially applied and then 20 µL of 1% Triton X-100 were added followed by two more pulses. Finally, the suspension was centrifuged at 4226× *g* for 10 min. Protein concentration was determined using the Pierce®BCA Protein Assay Kit (Thermo Scientific) as recommended by the manufacturer. Then, samples (30 µg of protein each) were separated by SDS-PAGE (12%) under denaturing conditions. In this assay α-tubulin was used as loading-control. Gels were transferred to PVDF membranes for 1.5 h at 300 mAmp at 10 °C, after which the membranes were blocked with 5% (*w*/*v*) non-fat dried milk for 1 h at room temperature (24 °C) under constant agitation. Membranes were washed three times with TBS-T 0.1% and incubated with either IG3 monoclonal antibody (1:500) in 5% non-fat dried milk/TBS–T 0.1% overnight at 4 °C or with monoclonal anti-α-tubulin antibody (TAT1 control; 1:15,000; kindly donated by Dr Keith Gull from Oxford University, UK) for 12 h. Thereafter, the probed membranes were washed three times for 1 h at 24 °C and then incubated with 1: 10,000 goat anti-mouse IgG coupled to HRP in TBS-T as secondary antibody (GE-Healthcare). Membranes were reacted with ECL Plus Lightning^®^ Western kit (Perkin Elmer Inc., Waltham, MA, USA) and photographed using a UVP High Performance Ultraviolet Transilluminator fitted with Launch DocItLS software. 

For zymograms, SDS-PAGE was performed using 12% polyacrylamide gels co-polymerized with 0.2% gelatin (Sigma); 30 μL of each total protein extracts under reducing conditions were loaded per lane and proteins were separated at 100 V for 2 h. The gels were further incubated in water containing 2.5% (*v*/*v*) Triton X-100 for 1 h and then washed 3 times in distilled water. To identify bands with proteolytic activity, the gels were incubated for 16 h at 37 °C in acetate buffer (pH 5.5) containing 1 mM DTT (constant agitation) and then washed 3 times with distilled water. Gels were stained with Coomassie blue (BioRad) for 2 h at 37 °C, washed with distilled water and incubated in 5% acetic acid /40% methanol (*v*/*v*). Gels were photographed as for Western blots.

### 4.6. Infection of Jirds and Faecal Microbiota Transplantation (FMT)

Animals were experimentally infected by gavage with 2.5 × 10^6^ of WB^−^ or the same number of WB^+^ or WB giardipain-pAC trophozoites in 500 μL of phosphate-buffered saline (PBS). For transplantation, fresh faeces were collected from jirds infected with WB giardipain-pAC-transfected trophozoites. Pooled faeces were homogenized in PBS and diluted to 0.5 g/mL, and 150 μL of suspension were given by gavage to the of recipient jirds. Three independent experiments were done using the different Giardia cell lines employing three animals per group (including uninfected controls). 

### 4.7. Monitoring the Number of Trophozoites in the Small Intestine of Jirds Infected with WB^−^ or with WB^+^ or with WB Giardipain-pAC Trophozoites

Jirds infected with either WB^−^, WB^+^ or WB giardipain-pAC were euthanized at 14 days post infection, and the entire small intestines obtained from each animal were removed and placed in PBS. Then, the intestinal tissues were incubated for 45 min in cold PBS with constant agitation to dislodge the trophozoites from the intestinal mucosa. Trophozoites in the suspension were pelleted by centrifugation at 700× *g* for 5 min, and the pellet was washed twice with PBS. Finally, the pellet was resuspended in 20 µl of cold PBS, and the trophozoites were counted in a haemocytometer.

### 4.8. Histopathological Examination

Jirds were euthanized and the duodenum was removed and “Swiss rolls” prepared [61]. For immunofluorescence, the intestinal tissue was embedded in optimal cutting temperature (OCT) medium (Tissue-Tek) and snap frozen at −70 °C until use. For immunohistochemistry, the tissue was immersed in 4% formalin (buffered with PBS; pH 7.2) and paraffin-embedded sections (3 μm) were prepared. Conventional histology with haematoxylin-eosin (H&E) or Alcian blue/periodic acid–Schiff (AB/PAS) stained sections were scanned using a slide scanner (Scan Scope CS2, Leica, Nussloch, Germany) and analyzed using Aperio ImageScope software (Aperio). The lengths and widths of villi were measured and the number of goblet cells per villus counted in 50 fields using a 100× magnification. Histopathological changes were scored as follows: 1: minimal and focal inflammatory cell infiltrate in the mucosa, with an intact epithelial layer; 2: mild, diffuse inflammatory cell infiltrates in mucosa and submucosa. 3: moderate, diffuse inflammatory cell infiltrates in mucosa and submucosa, with distorted villous structure; 4: moderate, crypt hyperplasia, fusion villi and diffuse inflammatory cell infiltrates in mucosa and submucosa.

### 4.9. Immunofluorescence

Duodenal tissue sections were fixed in formalin for 15 min (Sigma-Aldrich; Merck KGaA, Darmstadt, Germany), blocked for 1 h at room temperature with 1% BSA (Sigma-Aldrich; Merck KGaA, Darmstadt, Germany) and incubated overnight with primary antibodies anti-MUC2 (1:500) (Santa Cruz, CA, USA), pan-cadherin (1:500) (Santa Cruz, CA, USA), anti-Ki67 (1:1000) (Cell Signaling, Danvers, MA, USA) or anti-pH3 (1:1000) (Cell Signaling, Danvers, Massachusetts, USA). Appropriate fluorophore-labelled secondary antibody (1:1000) and Hoechst 33,342 [1: 20,000 dilution (Life Technology, Carlsbad, CA, USA)] were incubated for 1 h at room temperature, and coverslips mounted with Vectashield (Vector Laboratories, Burlingame, CA, USA). TUNEL staining was performed using the In Situ Cell Death Detection Fluorescein kit (Roche Diagnostics, Rotkreuz, Switzerland), according to the manufacturer’s protocol. Images were collected using a confocal laser scanning microscope (TCS SP8, Leica, Wetzlar, Germany).

### 4.10. Immunohistochemistry

Deparaffinated and hydrated tissue was blocked for 5 min with 5% hydrogen peroxide, heated for antigen retrieval with Diva Decloaker (Biocare, Tijuana, Mexico) and blocked with Rodent Block R (Biocare, Mexico). Then, the sections were incubated with anti-CD3 (1:1000) (Biocare, Mexico) for 30 min at RT, and revealed with Betazoid DAB Chromogen and Betazoid DAB substrate buffer (Biocare, Mexico). Tissue sections were then incubated in haematoxylin, dehydrated and cover-slipped for microscopic examination.

### 4.11. Bacterial DNA Isolation and Illumina-Based Sequencing of 16S Amplicons

Stool samples were collected from each control and infected jirds. Genomic DNA was extracted from 100 mg faeces using a Pure LinkTM Microbiome DNA Purification Kit (Thermo Fisher Scientific, Waltham, MA, USA) according to instructions. DNA was stored at −20 °C until processing.

Amplicon sequencing libraries were prepared by two-step PCR; 16S nuclear ribosomal RNA gene amplicons were prepared for the Illumina Miseq^®^ sequencing (Part # 15044223 Rev. B) according to the standard protocol provided by Illumina^®^. First, bacterial 16S rDNA was amplified using primers S-D-Bact-0341-b-S-17/S-D-Bact-0785-a-A-21 (Klindworth et al., 2013) in triplicate, pooled, purified using the QIAquick PCR purification kit (Qiagen, Hilden, Germany), and quantified using a Qubit 2.0 dsDNA HS Assay Kit (Life Technologies, USA). PCR products were ‘normalized’ to 5 ng/µL and then barcoded with oligonucleotides matching those from the Nextera XT Index Kit (Illumina, SD, USA). Successful amplification of the target 464 bp amplicon, which spans the V3 and V4 regions of the 16S rRNA gene, was verified by electrophoresis in 0.8% agarose gels [62]. Paired-end 16S amplicons were quantified, normalized and pooled. Second, amplicons were sequenced using V2 MiSeq reagent kit (500—cycles); multiplexed libraries were adjusted to a final concentration of 12 picomolar (pM), and contained 30% PhiX Control V3 library (Illumina, SD, USA).

### 4.12. Inferring Microbial Operational Taxonomic Units (OTUs)

The 1,192,734 reads across the V3–V4 regions produced by Illumina sequencing were trimmed, quality filtered, and the potential chimeric reads removed and demultiplexed by eliminating low abundant operational taxonomic units (OTUs), 571 308 sequences were retained for subsequent analysis. Taxonomic assignment was done using the QIIME2 suite [63]. Taxonomic grouping was conducted using data in the GreenGenes database, with the sequences grouping at 99% similarity. Alpha- and beta-diversities were computed using the vegan R package [64]. 

The ordering of samples from different groups was based, with (dis)similarities visualized by non-metric distance scaling (NMDS) and a permutational multivariate analysis of variance (PERMANOVA) using distance matrices (vegan:adonis) was performed. Alpha-diversity parameters and the occurrence of bacterial families were tested for all groups using a non-parametric Kruskal–Wallis test, adjusted for multiple comparisons by the Benjamini–Hochberg method using DESeq tool. For this analysis, in the case of a significant difference (*p* < 0.05), pairwise group comparisons (infected versus control) were performed using the Mann–Whitney test corrected for multiple testing, resulting in the indicated *p* values. From the results from DESeq2, heat maps were prepared to display changes in mean abundance (log2 fold changes or logarithm base two of the change) for the different clades, and comparisons between/among groups. Each row of the heat maps corresponded to a clade and each column to a comparison between conditions.

### 4.13. Statistical Analysis

Data were expressed as a mean ± standard error of the mean (SEM); comparisons were made between groups using a one-way ANOVA, with a Tukey’s post-hoc test, where *p* < 0.05 was considered significant. Statistical comparisons were made in GraphPad Prism 5.0 for Windows (San Diego, CA, USA). 

## Figures and Tables

**Figure 1 ijms-23-13649-f001:**
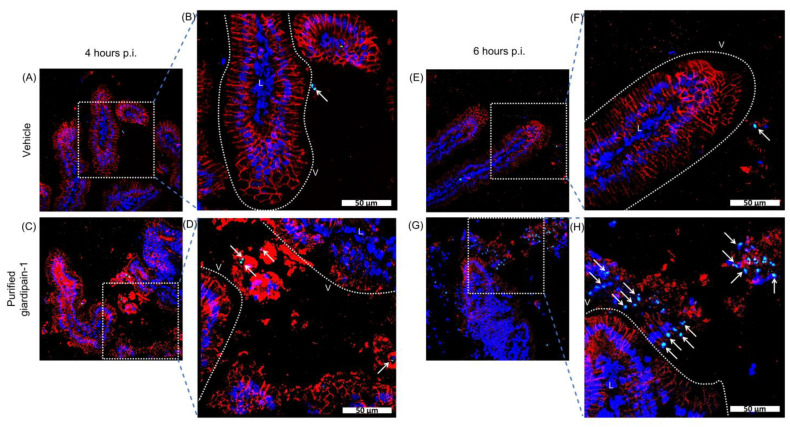
Giardiapain-1 induces apoptosis in intestinal epithelial cells of the jird duodenum. Staining by terminal deoxynucleotidyl transferase dUTP nick end labeling (TUNEL; green) of duodenal cryosections from control and giardipain-1 treated animals. Representative histological sections of the duodenum of control jirds or infected animals are included. The panels are identified as follows, sections from: (**A**) and (**B**) (amplification) control animals, (**C**) and (**D**) (amplification) giardipain-1 treated animals at 4 h p.i. (**E**) and (**F**) (amplification) control animals, (**G**) and (**H**) (magnification) animals treated with giardipain-1 at 6 h p.i. PAN-cadherin an intestinal epithelial cell marker = red. Nuclei = Blue. Dashed line delimits the villi. V = villi. LP = lamina propria. Arrows identify apoptotic epithelial cells that are extruded from the villi. Scale bar = 50 μm.

**Figure 2 ijms-23-13649-f002:**
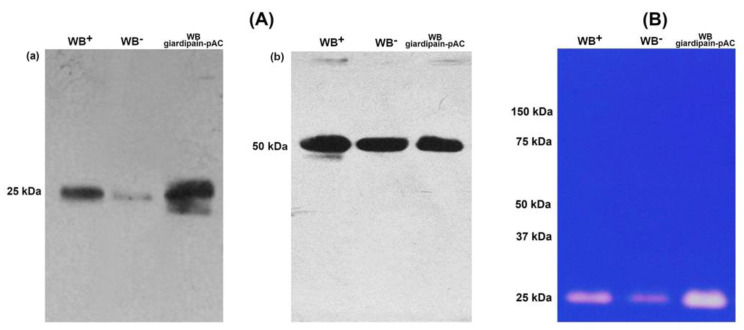
Recognition of giardipain-1 by monoclonal antibody 1G3 in total extracts of *G. duodenalis* WB giardipain pAC trophozoites, WB^+^ and WB^−^ and detection of its proteolytic activity. (**A**) Total extracts of *G. duodenalis* trophozoites were separated by SDS-PAGE transferred to PVDF membranes and probed with mAb 1G3. (a) A 25 kDa band was recognized with high (WB giardiapin pAC), wild type (WB^+^) or low expression (WB^−^) of giardipain-1 by 1G3 antibodies. (b) In these assays Tubulin was used as loading control. (**B**) Proteolytic activity detected in the same parasite samples. A specific proteolytic band with molecular weight of 25 kDa was detected with high (WB giardipain-pAC) wild type (WB^+^) and low (WB^−^) proteolytic activity.

**Figure 3 ijms-23-13649-f003:**
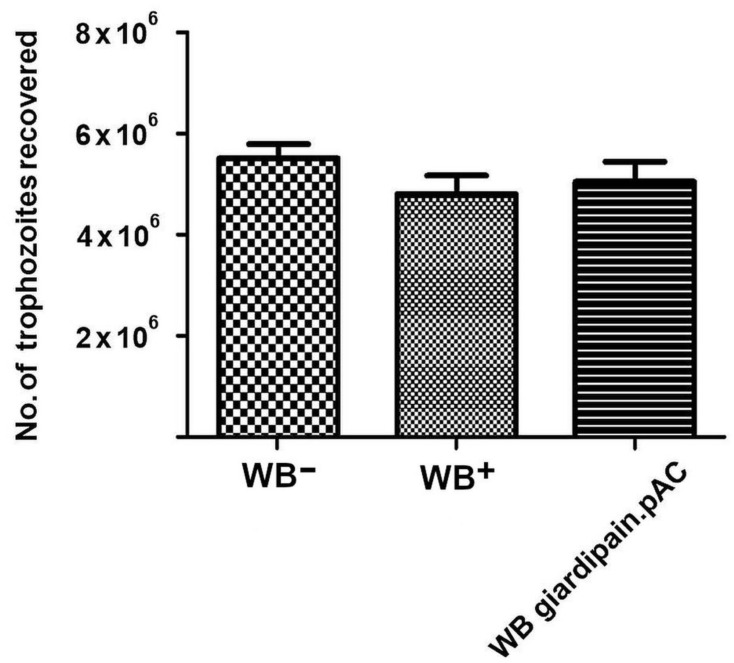
WB^+^, WB^−^ and WB giardipain-pAC trophozoites recovered from the duodenum of infected jirds at 14 days post-inoculation. Bars indicate the average number of trophozoites recovered from groups of three animals each ±SEM.

**Figure 4 ijms-23-13649-f004:**
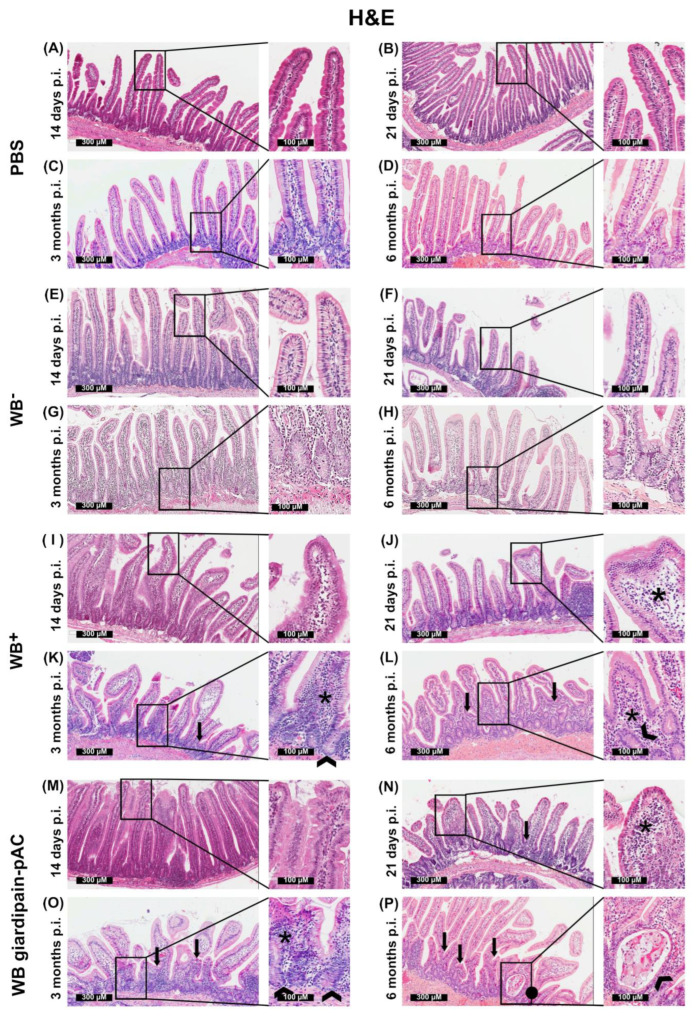
The infection of jirds with *Giardia duodenalis* trophozoites triggers microscopic alterations in the duodenum of these animals. Representative histological sections of the duodenum of control jirds or animals that were infected with WB^−^, or WB^+^ or WB giardipain-pAC. Time p.i. are 14 days, 21 days, 3 months and 6 months. Panels A to D are sections of control animals displaying normal mucosa with no alterations. In Panels E to H are sections of the duodenum from jirds infected with WB^−^ trophozoites, in panels I to L are section of duodenum from jirds infected with WB^+^ trophozoites and in panels M to P are sections of jirds infected with WB giardipain-pAC trophozoites. Hypertrophy, shortening and fusion of villi (black arrow) as well as hyperplasia of the intestinal crypts (head arrow), oedema and inflammatory infiltrate (*) were observed in WB^+^ infected jirds. The alterations were more severe at 6 months p.i. WB giardipain-pAC trophozoite induced similar histopathological alterations to the ones described for WB^+^ trophozoites However, the changes were more severe and cryptitis (black ball) was observed at 6 months p.i. A lower number of globet cells were observed in WB^−^ infected jirds. H&E stain, 300 μm and 100 μm scale bar.

**Figure 5 ijms-23-13649-f005:**
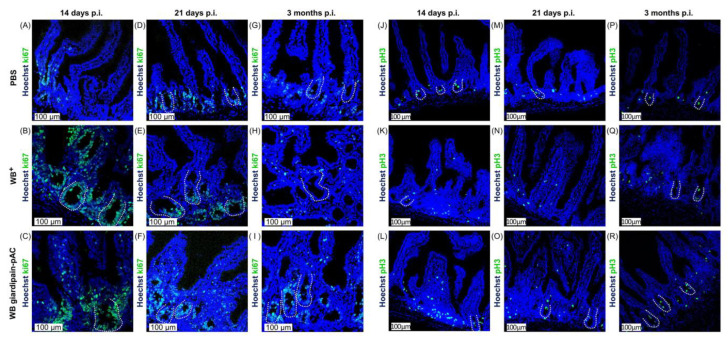
The infection of jirds with *Giardia duodenalis* trophozoites increases proliferation in the intestinal crypts of the duodenum in these animals. Proliferation in the duodenum of jirds that were infected with WB giardipain-pAC or WB^+^ trophozoites as assessed by the detection of Ki-67 (green) or of pH3 (green). Representative histological sections of the duodenum of control jirds or infected animals are included, and panels are as follows, sections stained with Ki-67 from: (**A**) control jirds, (**B**) animals infected with WB^+^ trophozoites (**C**) jirds infected with WB giardipain-pAC trophozoites, at 14 days p.i. (**D**) control animals (**E**) animals infected with WB^+^ trophozoites. (**F**) jirds infected with WB giardipain–pAC trophozoites, at 21 days p.i. (**G**) control animals (**H**) animals infected with WB^+^ trophozoites, (**I**) animals infected with WB giardipain-pAC trophozoites, at 3 months p.i. Sections stained with pH3 from: (**J**) control animals (**K**) animals infected with WB^+^ trophozoites, (**L**) animals infected with WB giardipain-pAC trophozoites, at 14 days p.i. (**M**) control animals (**N**) animals infected with WB^+^ trophozoites (**O**) jirds infected with WB giardipain-pAC trophozoites, at 21 days p.i. (**P**) control animals (**Q**) animals infected with WB^+^ trophozoites (**R**) animals infected with WB giardipain-pAC trophozoites, at 3 months p.i. Nuclei = blue. Dashed line marks the crypt border. Scale bar = 100 μm.

**Figure 6 ijms-23-13649-f006:**
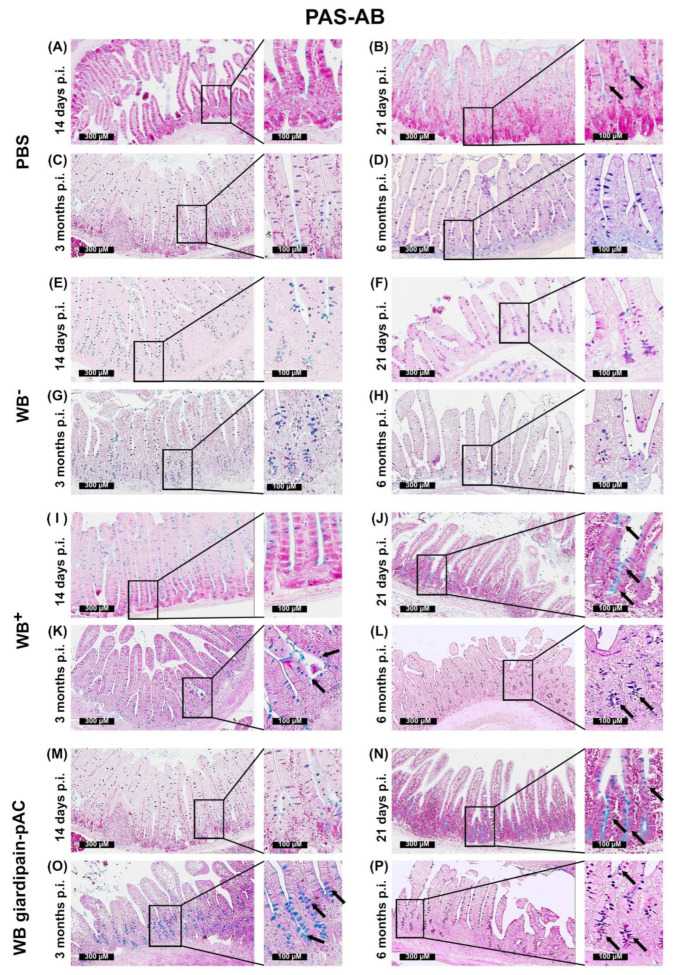
The infection of jirds with *Giardia duodenalis* trophozoites affects the biogenesis of goblet cells in the intestinal crypts of the duodenum in these animals. Goblet cells were identified by Alcian blue and Periodic acid–Schiff (PAS) staining of the duodenum of control jirds of animals that were infected with WB giardipain-pAC or WB^+^ or WB^−^ trophozoites. Representative histological sections of the duodenum of control jirds or infected animals are included. Times p.i. are 14 days, 21 days, 3 months and 6 months. PAS-AB + IEC representing the normal population of goblet cells in intestinal villi is observed in the control animals (**A–D**). PAS-AB ^+^ IEC increased in sections of the duodenum of the group infected with trophozoites. Panels are as follows: WB^-^ trophozoites (**E**) 14 days p.i. (**F**) 21 days p.i. (**G**) 3 months p.i. (**H**) 6 months p.i. WB^+^ (**I**) 14 days p.i. (**J**) 21 days p.i. (**K**) 3 months p.i. (**L**) 6 months p.i. and WB giardipain-pAC trophozoites (**M**) 14 days p.i. (**N**) 21 days p.i. (**O**) 3 months p.i. (**P**) 6 months p.i. At 14 and 21 days, goblet cells were detected in the crypt base. At 3 and 6 months, goblet cells were observed along the whole crypt-villus axis. A lower number of globet cells were observed in WB^−^ infected jirds. AB^+^ cells = blue color. PAS^+^ = purple color. Arrows indicate positively stained goblet cells.

**Figure 7 ijms-23-13649-f007:**
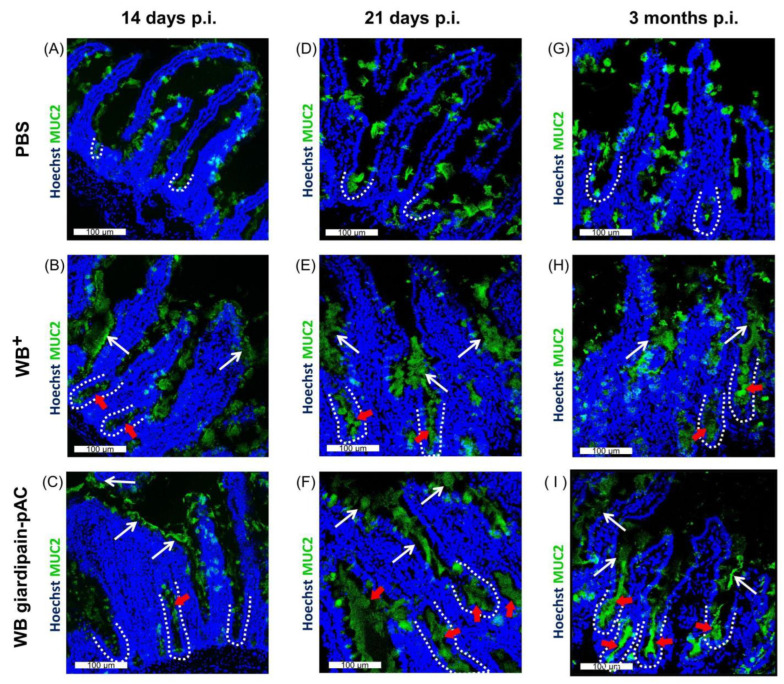
MUC2 is increased in the duodenum of jirds infected with *Giardia duodenalis* trophozoites. Mucin 2 (MUC2) was detected by immunofluorescence staining of the duodenum of jirds that were infected with WB giardipain-pAC or WB^+^ trophozoites. Representative histological sections of the duodenum of control jirds or infected animals are included. Panels are as follows, sections from: (**A**) control animals (**B**) animals infected with WB^+^ trophozoites (**C**) animals infected with WB giardipain-pAC trophozoites at 14 days p.i. (**D**) control animals (**E**) animals infected with WB^+^ trophozoites (**F**) animals infected with WB giardipain-pAC trophozoites, at 21 days p.i. (**G**) control animals (**H**) animals infected with WB^+^ trophozoites (**I**), animals infected with WB giardipain-pAC trophozoites, at 3 months p.i. MUC2 = Green. Nuclei = blue. Dashed line marks the crypt border. White arrow indicates secreted MUC2. Red arrow = indicates intracellular accumulation of MUC2. Scale bar = 100 μm.

**Figure 8 ijms-23-13649-f008:**
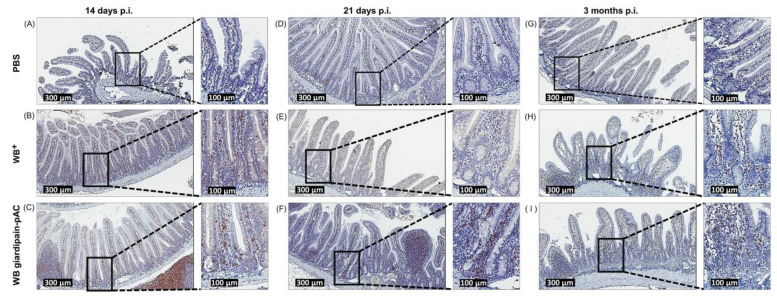
The infection of jirds with *Giardia duodenalis* trophozoites triggers the recruitment of CD3+ at the duodenum of jirds. CD3^+^ cells were determined by immunohistochemistry in histological sections of the duodenum of jirds that were infected with WB giardipain-pAC or WB^+^ trophozoites. Representative histological sections of the duodenum of control jirds or infected animals are included. Panels are identified as follows, sections from: (**A**) control animals, (**B**) animals infected with WB^+^ trophozoites, (**C**) animals infected with WB giardipain-pAC trophozoites at 14 days p.i. (**D**) control animals (**E**) animals infected with WB^+^ trophozoites (**F**) animals infected with WB giardipain-pAc trophozoites at 21 days p.i. (**G**) control animals (**H**) animals infected with WB^+^ trophozoites. (**I**) jirds infected with WB giardipain-pAc trophozoites at 3 months p.i. CD3^+^ cells are stained in brown. Scale bar = 300 μm and 100 μM.

**Figure 9 ijms-23-13649-f009:**
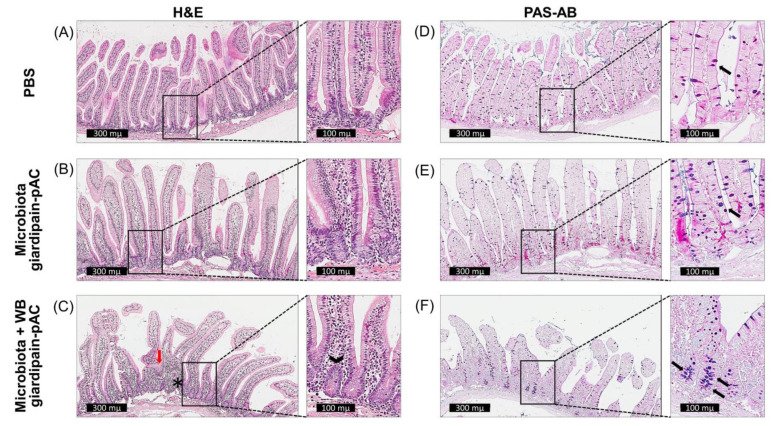
Microscopic alterations in sections of the duodenum of the animals treated with fecotransplants, then infected with WB giardipain-pAC trophozoites and euthanized at 21 days p.i. Representative histological and histochemistry sections of the duodenum of jirds that received the microbiota of previously infected jirds and then were infected with WB giardipain-pAC. Sections are from: (**A**) control animals displaying normal mucosa with no alterations. (**B**) jirds given microbiota from animals infected with WB giardipain-pAC trophozoites; in these animals slight thickening of villi was observed, (**C**) animals that received microbiota and that were infected with WB giardipain-pAC. In these atrophy, fusion of villi (red arrow) as well as hyperplasia of the intestinal crypts (head arrow) and inflammatory infiltrate (*) were observed by H&E stain, 300 μm and 100 scale bar. PAS-AB+ stained of sections from: (**D**) control animals in which the normal population of goblet cells in intestinal villi was observed. (**E**) jirds that were given microbiota from animals previously infected with WB giardipain-pAC trophozoites; in these sections a slight increase in goblet cells was observed and (**F**) animals given microbiota and then infected with WB giardipain-pAC. In these a marked increase in goblet cells was observed = arrow. Scale bar = 300 μm and 100 μM.

**Figure 10 ijms-23-13649-f010:**
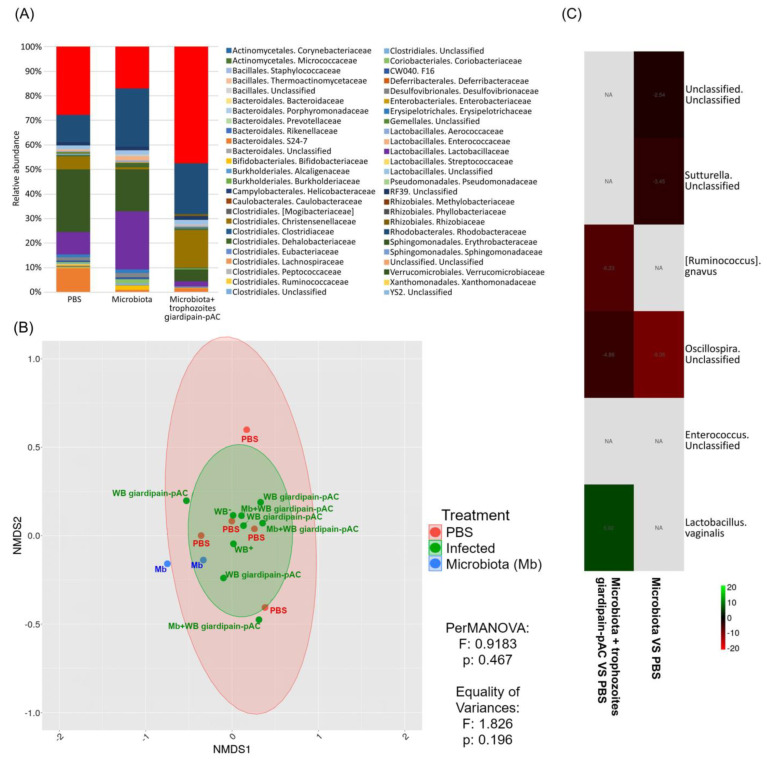
Determination of changes in the microbiota (Mb) composition of jirds that received different treatments and were euthanized at 21 days p.i. Composition of the microbiota in jird feces that received different treatments was determined by 16S ribosomal RNA gene amplicon sequencing. The data were compared between the control group (PBS) and the group of animals that was given microbiota from giardipain-pAC infected animals and then were infected with WB giardipain-pAC trophozoites. (**A**) Average relative microbial abundance at the family levels in the different groups. (**B**) Variation in beta-diversity of jird gut bacterial communities based on Bray–Curtis dissimilarities. (**C**) Heat map for relative abundance at species level with significant changes.

## Data Availability

The original contributions presented in the study are included in the article and Appendix A. Further enquiries can be directed to the corresponding author.

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
