# Peer review of "The Cysteine Protease Giardipain-1 from Giardia duodenalis Contributes to a Disruption of Intestinal Homeostasis"

_ijms, 2022, doi:10.3390/ijms232113649_

Round 1
Reviewer 1 Report
The manuscript is well-written. The study design is good and the conclusions are consistent with the results presented. The images are clear, well described and easy to understand.
I suggest a minor English language and style spell check
Author Response
Thank you very much for a thorough evaluation of our manuscript, please find our reply in the attachment.

Reviewer 2 Report
Manuscript IJMS-1902726 reports that purified Giardipain-1, a cysteine peptidase of Giardia duodenalis induces apoptosis and extrusion of epithelial cells at the tips of vili in infected jirds, thereby contributing to the pathogenesis of giardiasis. An infection model in jirds was used for experiments with 3 different strains /trophozoite cultures, G. duodenalis WB+ (a wt strain), WB- (WB trophozoites expressing a low amount of Giardipain-1) and WB giardipain-pAC (trophozoites with an overexpression of Giardipain-1). The manuscript presents nice set of experiments with very good histology. However, before it is published it is essential that authors explain how was “WB-“strain derived, as it is completely absent from the paper. It is an essential negative control. Furthermore, use of “WB-“should be mentioned in the abstract, to be able to write a sentence with a key role of Giardipain-1 in the pathogenesis of giardiasis. Otherwise one can say an important role the most.
Minor points:
Figure 1 legend, term “amplification” (lines 79, 80, 81) is inappropriate it should be magnification, or blow-up
Table 1 is unclear, especially in vitro part. The question is whether one needs the table at all. Couldn’t authors describe this in the text.
Figures 5, 8, 9 labels are too small to read.
Author Response

(The authors gave the same response as above.)
